# The Small Heat Shock Protein, HSPB1, Interacts with and Modulates the Physical Structure of Membranes

**DOI:** 10.3390/ijms23137317

**Published:** 2022-06-30

**Authors:** Balint Csoboz, Imre Gombos, Zoltán Kóta, Barbara Dukic, Éva Klement, Vanda Varga-Zsíros, Zoltán Lipinszki, Tibor Páli, László Vígh, Zsolt Török

**Affiliations:** 1Institute of Biochemistry, Biological Research Centre, 6726 Szeged, Hungary; bcsoboz@gmail.com (B.C.); gombos.imre@brc.hu (I.G.); dukic.barbara@brc.hu (B.D.); klement.eva@brc.hu (É.K.); zsiros.vanda@brc.hu (V.V.-Z.); lipinszki.zoltan@brc.hu (Z.L.); vigh.laszlo@brc.hu (L.V.); 2Institute of Medical Biology, University of Tromsø, 9008 Tromsø, Norway; 3Institute of Biophysics, Biological Research Centre, 6726 Szeged, Hungary; kota.zoltan@brc.hu (Z.K.); pali.tibor@brc.hu (T.P.); 4Single Cell Omics Advanced Core Facility, Hungarian Centre of Excellence for Molecular Medicine, 6726 Szeged, Hungary

**Keywords:** small HSP, lipid–protein interaction, membrane chaperone, membrane fluidity, stress response

## Abstract

Small heat shock proteins (sHSPs) have been demonstrated to interact with lipids and modulate the physical state of membranes across species. Through these interactions, sHSPs contribute to the maintenance of membrane integrity. HSPB1 is a major sHSP in mammals, but its lipid interaction profile has so far been unexplored. In this study, we characterized the interaction between HSPB1 and phospholipids. HSPB1 not only associated with membranes via membrane-forming lipids, but also showed a strong affinity towards highly fluid membranes. It participated in the modulation of the physical properties of the interacting membranes by altering rotational and lateral lipid mobility. In addition, the in vivo expression of HSPB1 greatly affected the phase behavior of the plasma membrane under membrane fluidizing stress conditions. In light of our current findings, we propose a new function for HSPB1 as a membrane chaperone.

## 1. Introduction

The heat shock response is a highly conserved response of a cell to challenging environmental stress conditions [1], and the conserved set of proteins termed heat shock proteins (HSPs) is critical to the maintenance of cellular integrity under stress conditions. HSPs are primarily responsible for sustaining cellular protein homeostasis via their chaperone activity, by aiding the assembly and folding of proteins, and by inducing their degradation after irreversible damage [2].

Aside from these well-characterized functions, HSPs were also found to interact with biological membranes through binding to membrane lipids [3]. Based on previous observations, the interaction between HSPs and lipids could alleviate the deleterious consequences of stress on membranes and their integral proteins by stabilizing membrane structure [4]. Experimental evidence accumulating over the past several decades suggests that among HSPs, the family of small heat shock proteins (sHSPs) could specifically associate with lipids and membranes [5]. The sHSP family is considered the most functionally diverse of the HSPs, consisting of proteins homologous to each other mostly in their alpha-crystallin domain, forming large hetero/homo-oligomeric complexes [6]. The initial characterization of sHSP interaction with lipids and their subsequent role in preserving the integrity of membranes via these interactions has been established by studying the HSP17 protein of the blue-green algae *Synechocystis* PCC 6803. The greater proportion of HSP17 was found to be associated with thylakoid membranes [7,8]. Its interaction with lipids increased the microviscosity of large unilamellar vesicles, consisting of synthetic or cyanobacterial lipids [9] and stabilized the lamellar liquid crystalline phase at the expense of the non-lamellar phase in membranes composed of the non-bilayer lipid, di-elaidoyl-phosphatidylethanolamine [10]. These studies [8,9,10] were among the first to demonstrate that HSP17 not only had an affinity towards specific membrane lipids, but also a previously unrecognized ability to stabilize membranes by modulating their lipid phase behavior. Several other examples, such as the sHSP proteins, IbpA and IbpB, of *Escherichia coli* [11] and HSP16 of *Mycobacterium tuberculosis* [12] reinforced the idea of the association of sHSPs with membranes. Moreover, the sHSP, Lo18, of the lactic acid bacterium, *Oenococcus oeni*, was upregulated and localized to the membrane fraction upon administration of the membrane fluidizer, benzyl alcohol (BA) [13]. Lo18 was also shown to interact with liposomes formed from lipids of *Oenococcus oeni* membranes and to reduce membrane fluidity of these vesicles at elevated temperatures [13]. Examples of the lipid interaction of sHSPs in other organisms are sparse. So far, only a few mammalian sHSPs were found to be associated with membranes. While HSP11 [14] and the muscle-specific HSPB2 [15] were linked to mitochondrial membranes, another mammalian sHSP, α-crystallin, was shown to associate with the plasma membranes of fiber cells in the lens of the eye [16] and to have a stabilizing effect on model membranes formed from synthetic lipids [10].

While it is clear that members of the sHSP family interact with membrane lipid vesicles in vitro, it is unclear whether this is a general property of sHSPs among different species and of different members of the sHSP family within a species. Moreover, detailed knowledge of how sHSPs affect membrane physical properties and whether they restore membrane functionality during/after heat shock is still lacking with only few reports. The existing studies suggest that the functional consequences of the association of sHSPs with membranes may include a reduced level of fluidity [9], elevated bilayer stability [10], and the overall restoration of membrane functionality during heat stress. Based on the above studies, we hypothesized that sHSPs may have a membrane-protective role across species to keep the membrane structure intact during elevated temperatures or other membrane-perturbing conditions. The association between sHSPs and membranes may constitute a general mechanism that preserves membrane integrity when the lipid order is compromised. One of the most ubiquitously expressed mammalian sHSPs, HSPB1, appears to be a potential candidate that could serve as a general membrane-stabilizing protein in mammalian cells. HSPB1’s membrane association was initially described in the context of its interaction with the membrane-connected cytoskeletal network [17,18,19], but recently, it has been reported to associate with synthetic liposome membranes [20]. Other physiological observations further substantiated the possible association between HSPB1 and the plasma membrane by demonstrating that HSPB1 was among the few HSPs that were upregulated upon the treatment of mouse cells with the membrane fluidizing compound, BA [21,22].

In this study, we aimed to describe the potential lipid/membrane interaction of HSPB1. We were particularly interested in determining if HSPB1 could affect the physical state of membranes and the mobility of lipids to a similar degree as described for other sHSPs. Overall, we aimed to find out if HSPB1 contributed to membrane integrity during and after stress conditions.

## 2. Results

### 2.1. HSPB1 Interacts with Lipids: A Monolayer Study

Initially, we investigated the interaction of HSPB1 with lipids by the Langmuir monolayer method. This technique allowed us to measure the surface pressure increase in the lipid monolayer caused by insertion of a chosen molecule between the lipid molecules spread on the air–water interface. HSPB1 showed a strong preference for a distinct group of phospholipids modeling biological membranes with different fluidity and phase behavior. The increase in surface pressure (Δπ), which is proportional to the number and level of proteins inserted into the monolayer, is concentration-dependent (Figure 1a). A protein concentration of 1 μM giving a significant surface pressure increase in all lipid mixtures was chosen for subsequent experiments. To test the effect of membrane fluidity, mono-component lipid monolayers of dioleoyl phosphatidylcholine (DOPC) or 1-palmitoyl-2-oleoylphosphatidylcholine (POPC) were applied. We observed stronger interaction with DOPC, which forms more fluid monolayers than POPC at a given initial surface pressure (Figure 1b). The elevation of surface pressure by protein insertion was reversed after proteinase K treatment (Figure 1d). Biomembranes contain raft microdomains enriched in phospholipids, sphingomyelin (SM), and cholesterol. To mimic the phase behavior of these microdomains, a ternary lipid mixture of POPC, SM, and dihydrocholesterol (DChol) (1:1:1) was applied, which contained both liquid-disordered fluid (Ld) and liquid-ordered (Lo) raft phases [23]. HSPB1 showed a weaker interaction with this raft mixture compared to POPC alone, while Δπ increased significantly when the amount of POPC was doubled in the mixture. Since ternary mixtures with higher PC ratios contain more liquid-disordered phases and higher overall lateral diffusion [23,24] (POPC/SM/DChol = 2:1:1) (Figure 1c), the significantly increased interaction of the protein with the lipids suggests a higher preference of HSPB1 for the liquid-disordered phase or a more fluid lipid phase, in general. This explanation was confirmed by the experiment where the interaction could be decreased by increasing the amount of the Lo phase by increasing the percentage of DChol in the mixture (Figure 1c); however, above a certain cholesterol level (2:1:3), the lipid interaction of HSPB1 was elevated again. The formation of free hydrophobic cholesterol patches in the monolayer at this high cholesterol concentration could account for this phenomenon [25]. These data suggest that HSPB1 tends to interact with lipids or lipid mixtures having higher fluidity, and the presence of free cholesterol could also be crucial for its binding to lipids. As the hydrophobic core of the bilayer becomes more accessible in the presence of free cholesterol [26], it is plausible that it can contribute to the binding of HSPB1 by creating a hydrophobic docking surface on the membrane.

### 2.2. HSPB1 Interaction Decreases Both Rotational and Lateral Fluidity in Model Membranes

In recent decades, spin-label EPR spectroscopy has become a useful technique for studying lipids, biological membranes, and lipid–protein interactions. We utilized this method to determine the effect of HSPB1 on the rotational mobility of lipids in model membranes.

The EPR spectra of 5-doxyl-stearic acid (5-SASL) and the corresponding outer splitting values are shown in Figure 2a,b, respectively, in the membranes of DOPC and POPC and in the absence and presence of HSPB1. The spectral parameter outer splitting (2A_max_) represents the mobility of the lipid acyl chain segments bearing the nitroxyl group. The spectra of the pure membranes are qualitatively similar, but their outer splitting reflect the expected difference in chain dynamics of phospholipids with one (POPC) vs. two (DOPC) double bonds; POPC has more restricted rotational dynamics (as evidenced by larger outer splitting) than DOPC, because of the disordering/fluidizing effect of lipid chain unsaturation in membranes. Adding HSPB1 to DOPC causes a reduction in membrane fluidity, indicating an interaction between the lipids and the protein, whereas it has a negligible effect on POPC membranes (Figure 2a,b). We also measured the same samples using 16-SASL spin labeling, in which the nitroxyl label came from the more hydrophobic central region of the membrane. Those spectra did not show any difference in the presence of HSPB1, suggesting that the interaction between lipids and HSPB1 occurs in the head group region of the membrane (data not shown).

The lateral diffusion of STAR488-PEG-cholesterol in the supported bilayer of DOPC was measured by image-based total internal reflection–fluorescence correlation spectroscopy (ITIR-FCS). The diffusion coefficient (D) reflects the lateral diffusion of the probe and can be calculated from the diffusion law graph [27]. The D value of the fluorescent probe in DOPC was 4 µm/s^2^, which is similar to values found in other fluid phase studies [28]. HSPB1 administration resulted in a 75% drop in D, suggesting drastic structural alterations in the DOPC bilayer induced by the HSPB1 interaction (Figure 3a). The reduced lateral mobility was almost completely restored by subsequent proteinase K treatment. Interestingly, the same HSPB1 treatment affected the mobility of the fluorescent cholesterol probe only slightly and in the opposite direction if the model membrane was made of DOPC/SM/DChol (1:1:1), modeling a liquid-ordered, membrane raft composition (Figure 3b). This is in line with our observation that heat shock-induced HSPB1 is partially present in the membranes of B16-F10 cells, and within the total membrane fraction, it predominantly accumulates in the non-raft membrane regions (Appendix A). Proteinase K treatment reduced the effect of HSPB1 on the D value in both supported bilayers.

### 2.3. The Presence of HSPB1 Preserves Membrane Order in Mammalian Cells 

After we characterized the fluidity-sensitive interaction of HSPB1 in model systems, we tested its effect on the membranes of living cells. We used *E. coli* bacteria and B16-F10 murine melanoma cells, which both overexpress HSPB1. Environment-sensitive, membrane-intercalating fluorescent probes are widely available and can be used to follow the alterations in membrane properties such as fluidity and lipid packing. One of these probes, di-4-ANEPPDHQ, shows an emission shift according to its localization in a liquid-ordered or liquid-disordered lipid phase. Using this probe allowed us to quantitate membrane order by determining the general polarization (GP). This ratiometric value was calculated from fluorescence intensities recorded in two spectral channels. As a normalized ratio, GP provides a measure of membrane order, in the range between −1 (liquid crystalline) and +1 (gel), where a lower GP value means higher membrane fluidity. First, we tested whether the overexpression of HSPB1 altered the GP of di-4-ANEPPDHQ, and found only slightly more ordered membranes in the bacteria used for HSPB1 production and purification, but no spectral shift in B16-F10 murine cells (Figure 4). This difference could be due to variations in the efficacy of HSPB1 expression in the two cell types (Appendix A). To perturb the membrane structure, we used the membrane-fluidizing agent, benzyl alcohol (BA). Exposure to BA significantly decreased the GP in B16-F10 cells, suggesting a lower membrane order (higher fluidity). However, the fluidizing effect of BA was completely blocked by prior HSPB1 overexpression in both bacterial and mammalian cells (Figure 4). The artificially increased expression of HSPB1 in B16-F10/HSPB1 cells was comparable to the levels induced by heat or BA treatment in wild-type cells (Appendix A).

## 3. Discussion

Biological membranes present a unique barrier, critical for the compartmentalization of a living cell from its environment. Factors compromising the structure of membranes could result in serious consequences for the integrity of a cell. Thus, characterization of the mechanisms for maintaining and repairing cell membrane structures is crucial for a deeper understanding of cellular physiology, stress responses, and the development of membrane-associated pathologies. It is presumed that membrane-interacting proteins, potentially early proto-HSPs, played an important role in the formation and stabilization of early membranes during evolution [29]. It is likely that this could be the origin of the documented membrane interactions for various HSPs [29]. HSPs are part of a cellular stress intervention pathway, and one of the positive results of these interactions could be the stabilization of membranes during stress [4,30].

In our study, we tried to widen the scope of understanding of the action of HSPs at the membrane level by describing the interaction between the stress protein, HSPB1, and lipid membranes in a fluidity-dependent manner. Our findings are consistent with the reported actions of other membrane-associated small HSPs. However, in contrast to the few reports on membrane-interacting mammalian sHSPs, the expression of HSPB1 is not tissue-specific. Thus, the action of this protein on membranes could be considered more as a ubiquitous and robust cellular tool for repair and maintenance of membranes. Based on our data, the range of intracellular functions of HSPB1 can now be extended to include membrane stabilization, through the lipid–protein interaction delineated here. The functional results of this interaction can potentially be manifold. For example, by directly stabilizing the membrane structure, HSPB1 could alleviate the deleterious effects of membrane over-fluidization, which can lead to the aggregation of membrane proteins even in the absence of heat stress.

The bilayer properties of membranes have an influence not only on the function of proteins embedded in them but can also be a defining factor in their capacity for aggregation within the membrane [31]. The fluid state of a membrane can affect the aggregation of normally soluble membrane-resident proteins. This has been indicated by a coarse-grained molecular dynamics simulation where the fluidity of the membrane was shown to have a determining effect on the rate of amyloid nucleation and subsequent aggregation, which was mainly caused by increased exposure to the hydrophobic core of the bilayer [32]. An indication that the size and curvature of a membrane are influencing factors in the amyloid fibrillation process has also been described in the highly fluid DOPC model membranes [33]. Another molecular simulation study demonstrated that increased membrane fluidity could be an important factor for α-synuclein membrane binding [34]. The overexpression of HSPB1 in a transgenic mouse model has also been found to decrease neuronal apoptosis upon membrane fluidizing ethanol treatment [35]. Overall, the stabilization of hyper-fluidic membranes by HSPB1 could have a major impact on the aggregation of membrane proteins. The direct interaction between HSPB1 and membrane proteins could represent an unorthodox chaperone activity separate from its classical function. The induction of HSPB1 has been connected with membrane lipid rafts and caveolae-1 signaling. HSPB1 levels were found to be lower in caveolin-1-deficient breast cancer cells [36], and the treatment of keratinocytes with agents such as filipin or methyl-β-cyclodextrin that disrupt lipid raft-caveolae suppressed sulfur mustard-induced HSPB1 mRNA and protein expression [37]. The alterations in the membrane cholesterol pool have been reported to have a fine-tuning effect on HSPB1 expression under stress conditions [38], suggesting that stress induction of HSPB1 can be partially regulated at the membrane level. For example, the surface coalescence of lipid rafts, which is a key event in lipid raft-associated signal initiation, is highly dependent on the fluid state of the membrane [39]. Thus, it is plausible that the readjustment of the membrane fluid state by HSPB1 could act as a re-setting mechanism, decreasing its expression as part of a negative feedback loop.

In our study, HSPB1 showed strong preference for DOPC as an interaction partner compared to POPC, and was more active in fluid ternary lipid mixtures. Our EPR measurements confirmed that HSPB1 operated in the lipid head group region, suggesting that, unlike HSPA1, it does not incorporate into the bilayer, but the interaction only occurs at the membrane surface. Our data show, however, that this interaction is sufficient for the protein to exert its effect on the physical structure of the bilayer by modulating the rotational mobility of the lipids. The active membrane modulatory nature of this interaction was verified by fluorescence correlation spectroscopy measurements. The presence of HSPB1 decreased the mobility of lipids forming the fluid phase in a supported bilayer, suggesting a general counteraction against imposed membrane fluidity. This effect was greatly reduced when cholesterol was present in the bilayer, which is in agreement with our observation (Appendix A) and with those in the literature [40] that described HSPB1 as only being present in the non-raft fraction of the membrane. HSPB1 was recently reported to interact with POPS, POPG, and POPC vesicles with higher affinity towards lipids with electrostatic charge (POPS and POPG). These experiments, which were performed with lipids containing the same acyl chains, are suggestive of the possibility that HSPB1 interaction with membranes depends on charged lipid head groups [20]. However, our biophysical measurements shed a different light on how critical the fluidity and packing order of lipids is in the lipid interaction of HSPB1. Based on our results, we hypothesize that HSPB1 interaction strongly depends on the phase properties of the interacting membrane, aside from the electrostatic charges of the lipid head groups, thus extending the former model. According to our observations, HSPB1 acts as a general membrane-stabilizing agent when the membrane fluidity increases and the hydrophobic regions of the membrane are increasingly exposed. In this regard, HSPB1 could be considered as a membrane-related stress-responsive actor that balances out the deleterious effect of sudden increases in membrane fluidity. HSPB1 modulates the membrane towards a more ordered state, as revealed by our experiments in which fluidization by BA was prevented by overexpression of the small heat shock protein in mammalian and bacterial cells. This feature of HSPB1 likely evolved under heat stress in order to compensate for the increased membrane fluidizing effect of heat. One could argue that the immediate effect of heat-induced membrane fluidization did not overlap with the stress-induced expression of this protein, as the former is immediate and the latter is a consequent event. We suggest that the membrane stabilizing function of HSPB1 represents an adaptive response of cells, allowing them to build up an acquired resistance to the next stress event. Therefore, we propose that stress-induced HSPB1 is a part of a toolkit for acquired stress resistance against membrane over-fluidization, and possibly against membrane-level stress conditions, in general. This hypothesis is in line with the widely documented literature on the ubiquitous role of HSPB1 in acquired stress resistance.

## 4. Materials and Methods

### 4.1. Expression and Purification of HSPB1

Recombinant human HSPB1 was expressed in *Escherichia coli* BL21 (DE3) (Thermo Fisher Scientific, Waltham, MA, USA) cells by using the pAK3038Hsp27 plasmid [41]. The subsequent purification of the recombinant protein was carried out as described in Buchner et al. [42]. In brief, *E. coli* BL21 (DE3) cells harboring the pAK3038Hsp27 plasmid were grown in the presence of ampicillin (Sigma-Aldrich, St. Louis, MO, USA) to the desired optical density, then induced with 0.5 mM isopropylthiogalactoside (IPTG) (Sigma-Aldrich). The expression of HSPB1 upon IPTG induction was assessed by Western blotting (Appendix A). Cells were incubated for 3 h upon IPTG induction, then harvested by centrifugation for 10 min at 2600× *g* and at 4 °C and lysed as described in [42]. After lysis, the proteins were precipitated with 35% ammonium sulfate and purified by ion exchange chromatography on a Fractogel EMD DEAE column (Merck Millipore, Burlington, MA, USA) using a 50 to 600 mM NaCl linear gradient. Eluted fractions were characterized for the presence of HSPB1 by SDS-PAGE and Western blot analysis (anti-HSPB1, SMC-161, StressMarq, Victoria, BC, Canada).

### 4.2. Langmuir Monolayer Experiments

Monolayer experiments were carried out essentially as described in [43] using a KSV3000 Langmuir-Blodgett instrument (KSV Instruments, Helsinki, Finland) and a Teflon dish containing 6.5 mL of PBS buffer with a surface area of 9 cm^2^ at 23 °C. Surface pressure was measured by the Wilhelmy method, using a platinum plate. Monomolecular lipid layers were spread on a buffer–air interface to give the desired initial surface pressure. The sub-phase was continuously stirred with a magnetic bar. Different concentrations of purified HSPB1, and subsequently, 1 µg/mL proteinase K (Thermo Fisher Scientific) was added underneath the monolayer with constant stirring.

### 4.3. EPR Spectroscopy

For spin-label EPR measurements, 5 μL of 5-SASL or 16-SASL spin label solution (Sigma-Aldrich) in ethanol (2 mg/mL) was added to 1 mg of lipid in chloroform solution (resulting in a ca. 50:1 lipid/spin label molar ratio). After vortexing, the solution was dried under nitrogen gas and incubated under vacuum overnight at room temperature. The dried samples were hydrated with PBS buffer (pH 7) above the melting temperature of the lipid. HSPB1 protein was added to the lipid suspension from an 11 mg/mL stock solution in PBS, resulting in a lipid:protein molar ratio of 100:1 in the samples.

A glass capillary with an internal diameter of 1 mm was then filled with 10 μL of spin-labeled sample. The EPR spectra were recorded using either a Bruker (Rheinstetten, Germany) ECS-106 or a Bruker ELEXSYS-II E580 X-band spectrometer at room temperature, with the following instrument settings: microwave frequency of 9.4 GHz, microwave power of 5 mW, field modulation of 0.7 G, scan range of 100 G, and conversion time of 40.96 s. Final spectra were the mean of four scans. Data analysis was performed using Igor Pro (Wavemetrics, Inc.; Portland, OR, USA).

### 4.4. ITIR-FCS

Giant unilamellar vesicles of DOPC or ternary lipid mixtures of DOPC/egg-sphingomyelin/cholesterol (Sigma-Aldrich) were prepared in 100 mM sucrose solution by electroswelling using a Nanion Vesicle Prep Pro device (Nanion, Munich, Germany). Vesicle suspensions in 250 µL of ultra-pure water were pipetted onto coverslips. After 20 min, the supported lipid bilayers were formed and the solution was replaced with 250 µL of ultra-pure water. Objective-type TIRF illumination was used to achieve the thinnest excited sample volume, with a high numerical aperture objective (alphaPlan-FLUAR100; Zeiss, Oberkochen, Germany). The data were acquired using a ProEM512 EMCCD camera (Princeton Instruments, Trenton, NJ, USA) with a 3 ms effective exposure time and 20 × 40 pixel acquisition area per measurement (pixel size 0.16 μm). The ImFCS plugin (https://www.dbs.nus.edu.sg/lab/BFL/imfcs_image_j_plugin.html, accessed on 16 May 2022) for ImageJ software was used for data evaluation (Rasband, W.S., ImageJ, U.S. National Institutes of Health, Bethesda, MD, USA, https://imagej.nih.gov/ij/, 1997–2018, accessed on 16 May 2022). The data analysis was performed as described earlier [44].

### 4.5. Imaging of di-4-ANEPPDHQ and Calculation of General Polarization (GP) 

*E. coli* BL21 (DE3) (Thermo Fisher Scientific) and B16-F10 mouse melanoma cells (ATCC) were used in these experiments. For inducible protein expression, the *E. coli* BL21 (DE3) cells were induced with IPTG as described earlier in Section 4 (Appendix A). In the case of the B16-F10 cells, HSPB1 was inserted into the pcDNA 4/TO (Thermo Fisher Scientific) expression vector, and the pcDNA6/TR vector (Thermo Fisher Scientific) was used as the source of the tetracycline repressor. The cells were co-transfected with both vectors using ExGen 500 (Thermo Fisher Scientific). Colonies were selected by the simultaneous addition of zeocin and blasticidine (both from Thermo Fisher Scientific). The expression of HSPB1 was induced by the addition of doxycycline hyclate (Sigma-Aldrich, D9891) to the cell culture media (2 µg/mL) for 24 h before the experiment. The expression of HSPB1 upon doxycycline induction, heat shock, and membrane fluidizing BA treatment was assessed by Western blot (Appendix A).

Bacterial and mammalian cells were labeled with the environment-sensitive, membrane-incorporating dye, di-4-ANEPPDHQ (Thermo Fisher Scientific), added to the growth medium at final concentrations of 5 μM and 1.5 μM, respectively. Following a 5 min incubation at RT, image acquisition was carried out on a Leica TCS SP5 confocal system. An argon ion laser at 488 nm was used for excitation, and detection ranges of PMTs were set to 500–580 nm and 620–720 nm for the two emission channels, respectively. The di-4-ANEPPDHQ data were typically displayed as pseudo-colored GP images. The GP values were calculated according to the following equation: GP = (*I*_(500-580) − G *I*_(620-750))/(*I*_(500-580) + G *I*_(620-750)) using an ImageJ macro. *I* represents the intensity in each pixel in the indicated spectral channel (numbers are in nm) and *G* represents the calibration factor, which compensates for the differences in the efficiency of collection in the two channels. Further image processing of segmenting cells and cell membranes was performed with CellProfiler [45] in combination with ilastik [46]. More than 1000 ROIs of membranes were segmented on at least 5 images per sample. GP values were read out and sorted into classes to calculate distribution. The Kolmogorov–Smirnov test was performed to analyze the equality of GP distributions in sample pairs using the MATLAB software (MathWorks, Natick, MA, USA).

HSPB1 expression in *E. coli* cells was induced with 0.5 µM IPTG (Sigma-Aldrich) 4 h before the measurements. Doxycycline hyclate (Sigma-Aldrich) was used to induce the expression of HSPB1 in B16F10 cells 24 h before the measurement. Both bacterial and mammalian cells were incubated with BA (Sigma-Aldrich) for 15 min prior to the measurement.

## Figures and Tables

**Figure 1 ijms-23-07317-f001:**
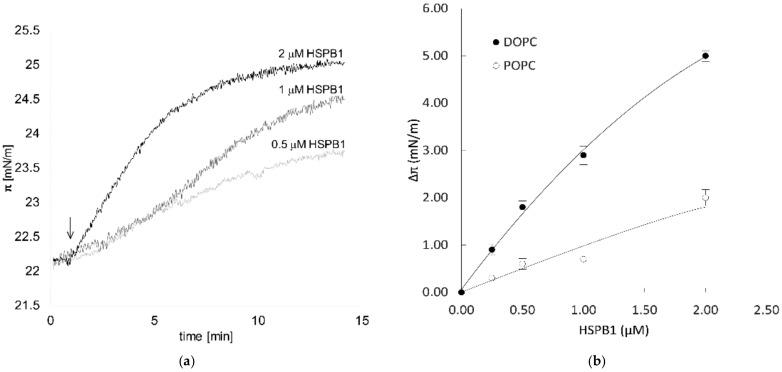
Lipid preference of HSPB1 measured by monolayer surface pressure. (**a**) HSPB1 was injected underneath the lipid monolayer of POPC and the surface pressure (π_i_) was measured after pressure equilibration. The black arrow indicates the time of injection. (**b**) Surface pressure change in DOPC or POPC monolayers five minutes following the addition of different amounts of HSPB1 underneath monolayers formed at an initial surface pressure of 22 mN/m. (**c**) Comparison of 1 µM HSPB1-induced maximum surface pressure increase in monolayers of pure POPC and different ternary lipid mixtures of POPC, SM, and DChol (π_I_ = 22 mN/m). (**d**) Effect of proteinase K treatment on the interaction of HSPB1 with POPC monolayers. Arrows indicate the time of HSPB1 (1 µM) and proteinase K (1 µg/mL) injection underneath the monolayer. Solid and dashed lines represent polynomial fitted curves.

**Figure 2 ijms-23-07317-f002:**
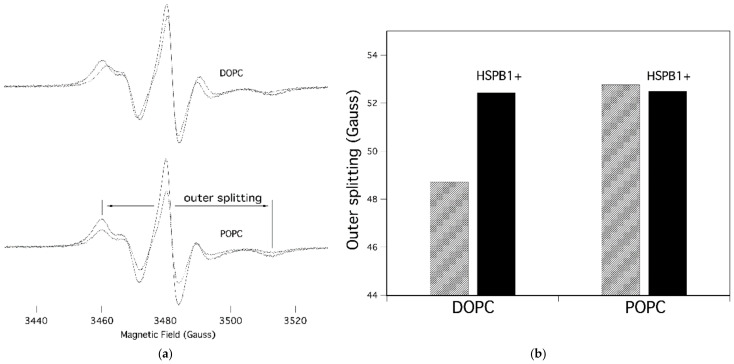
EPR spectra of 5-SASL in different lipid vesicles. (**a**) Spectra measured in the absence (dashed line) and in the presence (solid line) of HSPB1 (lipid to protein ratio, 100:1) are shown together. Spectral parameter outer splitting (2A_max_) is indicated. (**b**) Outer splitting (2A_max_) values of the control (striped) and HSPB1-containing (black) samples in different lipid membranes.

**Figure 3 ijms-23-07317-f003:**
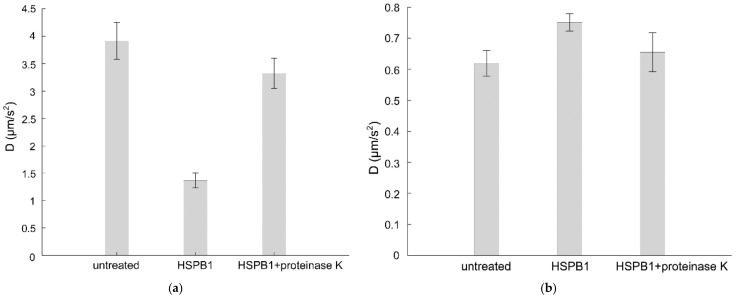
Effect of HSPB1 on the diffusion constant of STAR488-PEG-Chol fluorescent probe in supported bilayers. ITIR-FCS measurement was performed on a supported lipid bilayer made from (**a**) DOPC or (**b**) a 1:1:1 mixture of DOPC/SM/Chol before (first column) and after (second column) 15 min of 50 µg HSPB1 administration. The third column gives diffusion constants of HSPB1-treated samples after incubation with 1 µg/mL proteinase K for 15 min.

**Figure 4 ijms-23-07317-f004:**
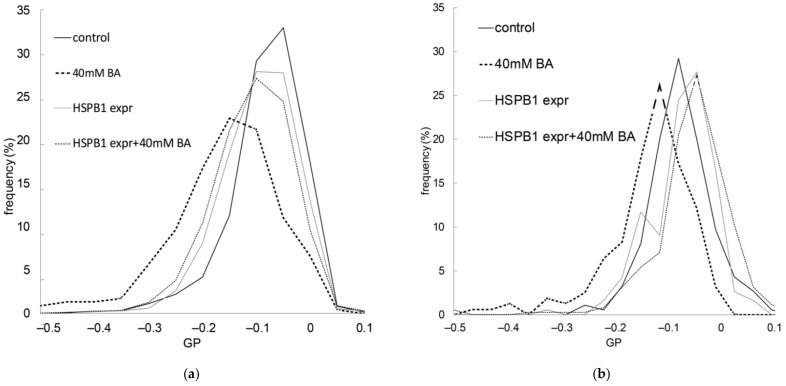
Effect of HSPB1 on membrane order impaired by benzyl alcohol (BA)**.** Membrane order was monitored by calculating the general polarization (GP) of the di-4-ANEPPDHQ environment-sensitive probe. Control and HSPB1-overexpressing (**a**) B16-F10 cells and (**b**) *E. coli* cells were pretreated with 40 mM BA for 15 min after HSPB1 expression. GP values were read out from processed and segmented fluorescence images and the distribution was plotted as a histogram. The Kolmogorov–Smirnov test was performed to analyze the equality of GP distributions in sample pairs. Control and BA-treated samples differed from each other significantly (*p* < 0.05), but GP distribution was equal in cells overexpressing HSPB1 with and without BA administration.

## Data Availability

The datasets generated and/or analyzed during this study are available from the corresponding authors on request.

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
