# Peer review of "The Small Heat Shock Protein, HSPB1, Interacts with and Modulates the Physical Structure of Membranes"

_ijms, 2022, doi:10.3390/ijms23137317_

Round 1

Reviewer 1 Report

The manuscript entitled "The small heat shock protein, HSPB1, interacts with and modulates the physical structure of membranes" by Balint Csoboz and colleagues studies the impact of HSPB1 on membrane fluidity in vitro, in bacterial and mammalian cells. The topic addressed in this study is of general interest and may have important implications to understand cell stress response and the mechanisms activated by the cells to maintain membrane integrity. Yet, the manuscript is written with a specific language that might prevent the general audience from appreciating its content and potential applications. A few suggestions to increase the general understanding are indicated below in the specific comments.

When reading the introduction and discussion the reader has the impression that the lipid chaperone function of HSPs is well-established. While a large body of evidence supports the chaperone activity of HSPs and sHSPs towards unfolded/misfolded proteins, the experimental evidence supporting their physical interaction with membranes in cells and their function in the stabilization of membrane integrity is still scarce. These aspects should be addressed, and some statements should be toned-down.

Concerning data interpretation, the authors apply BA or HS to the cells for 1 hr, let the cells to recover for 24 hrs and only then measure the impact of HSPB1 overexpression on membrane fluidity and lipid packing.  It is unclear why the authors did not measure the impact of HSPB1 overexpression immediately after HS and after a shorter recovery time. As mentioned in their manuscript, HSPB1 is a ubiquitous and abundant protein present in cells growing under resting conditions. Thus, it would be important to understand whether HSPB1 has a stabilizing effect on membrane fluidity immediately after acute stress. Depletion by siRNA of HSPB1 expression would also be useful to assess whether this has a negative impact on membrane fluidity after HS and BA treatment. These experiments will help to assess if HSPB1 has a prominent function in membrane stabilization also during acute stress and to what extent its function is dispensable (e.g. other sHSP might compensate in the absence of HSPB1 or not?).

Specific comments:

Introduction:

I suggest to tone-down this statement: "Overall, these studies provide evidence that members of the sHSP family interact with membrane lipids to influence the physical properties of these membranes. The functional consequence of the association of sHSPs with membranes include a reduced level of fluidity, elevated bilayer stability, and the overall restoration of membrane functionality during heat stress. "

While interaction of several sHSP with lipid vesicles in vitro has been clearly demonstrated, whether this is a general property of sHSP from different species and of different members of the sHSP family within a species is lacking. Moreover, detailed knowledge of how sHSP affect the membrane physical properties and whether they restore membrane functionality during/after heat shock is still lacking with only few reports existing.

Results:

Please, specify the name of the phospholipids used in this study rather than simply referring to their abbreviation, DOPC or POPC. For readers outside of the lipid vesicle field, it will be useful to explain shortly why the authors decided to use DOPC or POPC and in what these two types of lipid vesicles differ. The same comment applies to SM and Dchol: please, report the full name of the compounds and explain why they were added to the formulation. Please, highlight the physiological relevance of the lipid vesicles chosen for this study.

Figure 1a: for the general audience please briefly explain what the Langmuir monolayer method is and specify in the legend which type of lipid monolayer was used.

Figure 1b: Why did the author chose to use 1.0 uM HSPB1 for this assay? Does addition of 0.5 uM HSPB1 affect the surface pressure of DOPC or POPC or not? And is the effect incremental by addition of 2.0 uM HSPB1? Please explain why you selected 1.0 uM as optimal concentration for your studies.

Figure 1C: The authors use different combinations of lipids to mimic the raft organization of membranes and conclude that "HSPB1 tends to interact with lipids or lipid mixtures having higher fluidity". It is important to directly compare the fluidity of the different lipid compositions used in absence of HSPB1. To what extent do these formulations differ in terms of fluidity from one another?

Figure 2: the authors state that “The spectra measured in POPC show only a slight difference, which indicates a smaller effect on rotation mobility suggesting weaker lipid-protein interactions.” Could the author explain this difference? Is this correlating with a difference in membrane fluidity between DOCP and POPC?

Figure S1 and S2: "B16-F10 cells were subjected to heat shock at 42⁰C for one hour, then incubated for 24 hours." And “B16-F10 cells were subjected to heat shock at 42⁰C for one hour or left untreated, followed by an incubation for 24 hours.” Please rephrase for clarity; did the authors incubate the cells for 24 hours at 37⁰C? Why did the author analyze the distribution of HSPB1 in different membrane fractions 24 hrs after the recovery from stress? The authors should report the distribution of HSPB1 in different membrane fractions immediately after heat shock and also after a shorter recovery period (e.g. 2-4 hrs). It is difficult to interpret these results when taking into consideration only the 24 hr recovery time-point; this because the heat shock-induced changes in subcellular localization generally occur during the acute stress and are resolved within a few hours after the stress.

Please rephrase “heat shock-induced HSPB1 is present in the membrane of mammalian cells and predominantly accumulates in the non-raft membrane fractions”.

Finally, it would be good to include a positive control for another protein known to accumulate in the non-raft membrane fractions and, ideally, in the raft membrane fractions.

In their study the authors use a B16-F10 cell line (B16-F10/HSPB1) in which the overexpression of HSPB1 is induced upon addition of doxycycline (2 mg/ml). The description of the generation of this cell line should be reported in the material and method sections. The expression levels of HSPB1 in B16-F10 cells exposed to acute HS at 42⁰C and BA for 1 hour should also be shown. Was the protein already induced at this time-point? It is remarkable to observe such a strong HSPB1 induction 24 hrs after recovery from stress. Quantitation of the induction of HSPB1 expression upon acute HS and BA treatment and after recovery in B16-F10 cells and upon addition of doxycycline in B16-F10/HSPB1 cells should be included to directly compare the extent of induction in the different conditions (Figure S3B).

Figure 4 reports a distribution of GP values. (1) It is not clear how the distribution was calculated. How many images were analysed? The error bars in the histograms in the figure should be reported. (2) The detailed results of the statistical test to assess whether the distributions are different should be reported. (3) The characteristic values of GP for liquid-ordered and liquid-disordered lipid phases should be reported, otherwise it is difficult for the reader to understand the significance of the results. One would guess that, in order for the conclusions to be valid, these values should be close to -0.1 and -0.2, respectively. Is this the case?

Discussion:

Please tone-down statements such as “HSPs are part of a cellular stress intervention pathway, and one of the major positive results of these interactions is the stabilization of membranes during stress 4,28.”

The authors suggest that “HSPB1 could be considered as a membrane-related stress-responsive actor that balances out the deleterious effect of sudden increases in membrane fluidity.” The experiments were performed 24 hrs after exposure of the cells to either BA or HS. As mentioned-above, this cannot be considered as a “sudden” response and it would be important to understand whether HSPB1 plays similar effects on membrane fluidity immediately after acute HS and after a shorter recovery time.

Author Response

We would like to thank the reviewer for taking the time to assess our manuscript. We are certain that the feedback we got will improve the quality of our study. We have addressed all the points raised and our answers are as follows (in red):

General comments

When reading the introduction and discussion the reader has the impression that the lipid chaperone function of HSPs is well-established. While a large body of evidence supports the chaperone activity of HSPs and sHSPs towards unfolded/misfolded proteins, the experimental evidence supporting their physical interaction with membranes in cells and their function in the stabilization of membrane integrity is still scarce. These aspects should be addressed, and some statements should be toned-down.

Concerning data interpretation, the authors apply BA or HS to the cells for 1 hr, let the cells to recover for 24 hrs and only then measure the impact of HSPB1 overexpression on membrane fluidity and lipid packing.  It is unclear why the authors did not measure the impact of HSPB1 overexpression immediately after HS and after a shorter recovery time. As mentioned in their manuscript, HSPB1 is a ubiquitous and abundant protein present in cells growing under resting conditions. Thus, it would be important to understand whether HSPB1 has a stabilizing effect on membrane fluidity immediately after acute stress. Depletion by siRNA of HSPB1 expression would also be useful to assess whether this has a negative impact on membrane fluidity after HS and BA treatment. These experiments will help to assess if HSPB1 has a prominent function in membrane stabilization also during acute stress and to what extent its function is dispensable (e.g. other sHSP might compensate in the absence of HSPB1 or not?).

Response: We have revised the text according to the reviewer’s suggestions. We moderated our statements to reflect the scarcity of experimental evidence supporting the physical interaction of sHSPs with membranes and their function in the stabilization of membrane integrity.

Regarding the questions in connection with the basal expression of HSPB1, we agree with the reviewer that it would be important to study the effect of non-stress induced HSPB1 on the membrane. However, in the cells we used in the current study, we were not able to detect basal expression of this protein in normal growth conditions by western blot, and our mass spectrometry measurements showed only a non-significant amount of this protein. This was one of the reasons that we used this cell line for this study, since the lack of basal HSPB1 allowed us to assay the effect of the stress induced HSPB1 alone.

Nonetheless, we agree that it would be useful to study the effect of HSPB1 on the membrane upon stress with a system where it was abundantly expressed at a basal level. We tried to model this with artificial HSPB1 expression in B16-F10 cells (Fig. 4. and Fig. S3B). We believe that these results could serve well for follow-up studies where we will include a model with a basal high HSPB1 expression and compare it to an HSPB1-silenced version.

Specific comments

Point 1 (Introduction): I suggest to tone-down this statement: "Overall, these studies provide evidence that members of the sHSP family interact with membrane lipids to influence the physical properties of these membranes. The functional consequence of the association of sHSPs with membranes include a reduced level of fluidity, elevated bilayer stability, and the overall restoration of membrane functionality during heat stress. " While interaction of several sHSP with lipid vesicles in vitro has been clearly demonstrated, whether this is a general property of sHSP from different species and of different members of the sHSP family within a species is lacking. Moreover, detailed knowledge of how sHSP affect the membrane physical properties and whether they restore membrane functionality during/after heat shock is still lacking with only few reports existing.

Response 1: The introduction has been modified according to the reviewer’s suggestions, to include a more rational view of the existing data on sHSP-membrane interactions.

Point 2 (Results): Please, specify the name of the phospholipids used in this study rather than simply referring to their abbreviation, DOPC or POPC. For readers outside of the lipid vesicle field, it will be useful to explain shortly why the authors decided to use DOPC or POPC and in what these two types of lipid vesicles differ. The same comment applies to SM and Dchol: please, report the full name of the compounds and explain why they were added to the formulation. Please, highlight the physiological relevance of the lipid vesicles chosen for this study.

Response 2: The first paragraph of the Results section was rewritten according to the reviewer’s suggestion highlighting the purpose of the lipid compositions for studying the effects of membrane fluidity or lipid phase behavior on the interaction.

Point 3 (Results): Figure 1a: for the general audience please briefly explain what the Langmuir monolayer method is and specify in the legend which type of lipid monolayer was used.

Response 3:  An explanatory sentence is included in the revised version.

Point 4 (Results): Figure 1b: Why did the author chose to use 1.0 uM HSPB1 for this assay? Does addition of 0.5 uM HSPB1 affect the surface pressure of DOPC or POPC or not? And is the effect incremental by addition of 2.0 uM HSPB1? Please explain why you selected 1.0 uM as optimal concentration for your studies.

Response 4: After testing the effect of protein concentration on DOPC and POPC, a concentration of 1 mM giving a significant surface pressure increase in both lipid mixtures was chosen for all subsequent experiments. To answer the reviewer’s questions about whether 0.5 mM HSPB1 gave a measurable effect on both monolayers and if the addition of 2 mM protein had an additive effect, we included these results in a new figure (Fig. 1b).

Point 5 (Results): Figure 1C: The authors use different combinations of lipids to mimic the raft organization of membranes and conclude that "HSPB1 tends to interact with lipids or lipid mixtures having higher fluidity". It is important to directly compare the fluidity of the different lipid compositions used in absence of HSPB1. To what extent do these formulations differ in terms of fluidity from one another?

Response 5:  In the rewritten paragraph, we have included an explanation for the changes in the composition of the ternary mixtures referring to literature data.

Point 6 (Results): Figure 2: the authors state that “The spectra measured in POPC show only a slight difference, which indicates a smaller effect on rotation mobility suggesting weaker lipid-protein interactions.” Could the author explain this difference? Is this correlating with a difference in membrane fluidity between DOCP and POPC?

Response 6:  Thank you for this question, which prompted us to present our data more precisely. In order to make the differences and the effect clearer, we modified Fig. 2 and its legend, which now also shows the control values of the outer splittings (i.e., membranes in the absence of protein). The expected difference in the rotational dynamics between the two membranes is now pointed out in the text. HSPB1 clearly has an effect only on DOPC membranes.

Point 7 (Results): Figure S1 and S2: "B16-F10 cells were subjected to heat shock at 42⁰C for one hour, then incubated for 24 hours." And “B16-F10 cells were subjected to heat shock at 42⁰C for one hour or left untreated, followed by an incubation for 24 hours.” Please rephrase for clarity; did the authors incubate the cells for 24 hours at 37⁰C? Why did the author analyze the distribution of HSPB1 in different membrane fractions 24 hrs after the recovery from stress? The authors should report the distribution of HSPB1 in different membrane fractions immediately after heat shock and also after a shorter recovery period (e.g. 2-4 hrs). It is difficult to interpret these results when taking into consideration only the 24 hr recovery time-point; this because the heat shock-induced changes in subcellular localization generally occur during the acute stress and are resolved within a few hours after the stress.

Response 7: The cells were incubated for 24 hours at 37⁰C. The quoted sentence was rephrased for the sake of clarity. According to our experience, there were no detectable protein levels of HSPB1 in these cells under basal, non-stressed conditions for up to 2-3 hours following stress treatment. In our experiments, we aimed to model how this protein could contribute to membrane stabilization during subsequent stress. We agree that it would be interesting to study this phenomenon in a system that has basal HSPB1 expression. To test the ability of preexisting HSPB1 to protect membranes, we used E.coli and B16-F10 cells artificially expressing HSPB1 (Fig 4.). The fluidizing effect of benzyl alcohol was completely blocked by prior HSPB1 overexpression in both bacterial and mammalian cells. To emphasize this aim we modified the figure legend of Fig. 4. We agree with the reviewer that it would be meaningful to follow up on this finding by using a cell model intrinsically expressing HSPB1 under normal growth conditions using siRNA. We are aiming to include such experiments in our follow-up studies.

Point 8 (Results): Please rephrase “heat shock-induced HSPB1 is present in the membrane of mammalian cells and predominantly accumulates in the non-raft membrane fractions”.

Response 8:  The sentence has been rephrased more specifically reflect this finding.

Point 9 (Results): Finally, it would be good to include a positive control for another protein known to accumulate in the non-raft membrane fractions and, ideally, in the raft membrane fractions.

Response 9:  We have included the appropriate controls in the supplementary figures.

Point 10 (Results): In their study the authors use a B16-F10 cell line (B16-F10/HSPB1) in which the overexpression of HSPB1 is induced upon addition of doxycycline (2 mg/ml). The description of the generation of this cell line should be reported in the material and method sections. The expression levels of HSPB1 in B16-F10 cells exposed to acute HS at 42⁰C and BA for 1 hour should also be shown. Was the protein already induced at this time-point? It is remarkable to observe such a strong HSPB1 induction 24 hrs after recovery from stress. Quantitation of the induction of HSPB1 expression upon acute HS and BA treatment and after recovery in B16-F10 cells and upon addition of doxycycline in B16-F10/HSPB1 cells should be included to directly compare the extent of induction in the different conditions (Figure S3B).

Response 10:  The description of the generation of the HSPB1 expressing cell line is included in the revised Supplementary Materials and Methods. The expression of HSPB1 upon HS and BA treatment for 1 h is represented in Figure S3 in the supplementary section. The expression levels of HSPB1 in B16-F10 cells exposed to acute HS at 42⁰C and BA for 1 h was not detectable. In a previous study using a CHO cell line we characterized the induction of this sHSP as a function of heating time in detail. Up to 2 h, there was no induction detectable without recovery time at 37⁰C (Tiszlavicz et al., 2022, Fig.1.:  https://doi.org/10.3390/biomedicines10051172). We believe that the reason for such a strong HSPB1 expression 24 hours after stress stemmed from its role in acquired stress tolerance. During the recovery phase, the cells might try to build up resistance against subsequent stress exposure. In general, we agree on being as quantitative as possible, but in this special case, since HSPB1 levels under normal growth conditions cannot be detected in B16-F10 cells (Fig.S3), we believe that quantification wouldn’t reflect the induction value properly.

Point 11 (Results): Figure 4 reports a distribution of GP values. (1) It is not clear how the distribution was calculated. How many images were analysed? The error bars in the histograms in the figure should be reported. (2) The detailed results of the statistical test to assess whether the distributions are different should be reported. (3) The characteristic values of GP for liquid-ordered and liquid-disordered lipid phases should be reported, otherwise it is difficult for the reader to understand the significance of the results. One would guess that, in order for the conclusions to be valid, these values should be close to -0.1 and -0.2, respectively. Is this the case?

Response 11:  (1) The details of the calculation of GP distribution are included in the revised Materials and methods”. (2) To compare the distribution of datasets the two-samlple Kolmogorov-Smirnov statistical test was used (MATLAB software) to identify significant changes between histograms. (3) A more detailed description of GP values is included in the revised “Results” section.

Point 12 (Discussion): Please tone-down statements such as “HSPs are part of a cellular stress intervention pathway, and one of the major positive results of these interactions is the stabilization of membranes during stress 4,28.”

Response 12:  The statement in the revised discussion was modified according to the reviewer’s suggestion.

Point 13 (Discussion): The authors suggest that “HSPB1 could be considered as a membrane-related stress-responsive actor that balances out the deleterious effect of sudden increases in membrane fluidity.” The experiments were performed 24 hrs after exposure of the cells to either BA or HS. As mentioned-above, this cannot be considered as a “sudden” response and it would be important to understand whether HSPB1 plays similar effects on membrane fluidity immediately after acute HS and after a shorter recovery time.

Response 13:  We propose that HSPB1 may be involved in the adaptive response of the cell to protect it against further membrane-level stresses. The stress-induced HSPB1, by its membrane interaction, would protect cells from subsequent stress that could damage membrane structures and impair integrity. Therefore, we propose that (membrane-) stress-induced HSPB1 might be a part of a toolkit for acquired stress resistance against membrane over-fluidization and against membrane-level stress conditions, in general. We added this hypothesis to the revised discussion.

Reviewer 2 Report

In this study, the authors show that the heat shock protein hspb1 interacts with membranes and in particular with fluid membranes and preserve mammalian cell membrane by stabilizing it.

The study is interesting and clearly presented.

The authors may explain a little bit bette the langmuir monolayer method as it is important for the understanding of the study.

Author Response

We would like to thank the reviewer for their work and comments on the manuscript. Our responses are as follows:

Point 1: The authors may explain a little bit better the langmuir monolayer method as it is important for the understanding of the study.

Response 1:  Following the reviewer’s suggestion, the first paragraph of the Results section was rewritten to explain the technique and highlight the purpose of using the different lipid compositions.

Round 2

Reviewer 1 Report

The authors have adequately addressed all concerns raised and the manuscript has improved.